# Emergent constraints on TCR and ECS from simulated historical warming in CMIP5 and CMIP6 models

Femke J.M.M. Nijsse[1], Peter M. Cox[1], and Mark S. Williamson[1, 2]

[1]University of Exeter, College of Engineering, Mathematics and Physical Sciences, EX4 4QE, Exeter, UK
[2]University of Exeter, Global Systems Institute, EX4 4QE, Exeter, UK

**Correspondence:** Femke J.M.M. Nijsse (f.j.m.m.nijsse@exeter.ac.uk)

**Abstract.** Climate sensitivity to $CO_2$ remains the key uncertainty in projections of future climate change. Transient climate response (TCR) is the metric of temperature sensitivity that is most relevant to warming in the next few decades, and contributes the biggest uncertainty to estimates of the carbon budgets consistent with the Paris targets. Equilibrium climate sensitivity (ECS) is vital for understanding longer-term climate change and stabilization targets. In the IPCC 5th Assessment Report (AR5), the stated 'likely' ranges (16–84% confidence) of TCR (1.0–2.5 K) and ECS (1.5–4.5 K) were broadly consistent with the ensemble of CMIP5 Earth System Models (ESMs) available at the time. However, many of the latest CMIP6 ESMs have larger climate sensitivities, with 5 of 34 models having TCR values above 2.5 K, and an ensemble mean TCR of $2.0 \pm 0.4$ K. Even starker, 12 of 34 models have an ECS value above 4.5 K. On the face of it, these latest ESM results suggest that the IPCC likely ranges may need revising upwards, which would cast further doubt on the feasibility of the Paris targets.

Here we show that rather than increasing the uncertainty in climate sensitivity, the CMIP6 models help to constrain the likely range of TCR to 1.3–2.1 K, with a central estimate of 1.68 K. We reach this conclusion through an emergent constraint approach which relates the value of TCR linearly to the global warming from 1975 onwards. This is a period when the signal-to-noise ratio of the net radiative forcing increases strongly, so that uncertainties in aerosol forcing become progressively less problematic. We find a consistent emergent constraint on TCR when we apply the same method to CMIP5 models. Our constraints on TCR are in good agreement with other recent studies which analysed CMIP ensembles. The relationship between ECS and the post-1975 warming trend is less direct and also non-linear. However, we are able to derive a likely range of ECS of 1.9–3.4 K from the CMIP6 models by assuming an underlying emergent relationship based on a 2-box energy balance model. Despite some methodological differences, this is consistent with a previously-published ECS constraint derived from warming trends in CMIP5 models to 2005. Our results seem to be part of a growing consensus amongst studies that have applied the emergent constraint approach to different model ensembles and to different aspects of the record of global warming.

## 1 Introduction

The key uncertainty in projections of future climate change continues to be the sensitivity of global mean temperature to changes in the Earth's energy budget, called 'radiative forcing'. This sensitivity is usually characterised in terms of the global

mean temperature that would occur if the atmospheric carbon dioxide concentration was doubled, for which the radiative
forcing is reasonably well-known.

Two related quantities are used to characterise the climate sensitivity of Earth System Models (ESMs). Equilibrium climate
sensitivity (ECS) is an estimate of the eventual steady-state global warming at double $CO_2$. Transient climate response (TCR)
is the mean global warming predicted to occur around the time of doubling $CO_2$ in ESM runs for which atmospheric $CO_2$
concentration is prescribed to increase at 1% per year. Across an ensemble of ESMs, TCR values are less than ECS values
because of deep ocean heat uptake, which leads to a lag in the response of global temperature to the increasing $CO_2$ concentra-
tion (Hansen et al., 1985). The ratio of TCR over ECS tends to decrease with increasing ECS, and depends on spatial pattern
effects (Armour, 2017).

Despite decades of advances in climate science, the Earth's ECS and TCR remain uncertain. The 'likely' range of ECS (66%
confidence limit) has been quoted as 1.5 K to 4.5 K in all of the five Assessment Reports (ARs) of the Intergovernmental Panel
on Climate Change (IPCC) starting in 1990, aside from the fourth AR which moved the likely lower range temporarily to 2 K.
Similarly the likely range of TCR is given as 1 K to 2.5 K in the IPCC AR5, based on multiple lines of evidence.

There have been numerous attempts to constrain ECS using the record of historical warming or palaeoclimate data (Knutti
et al., 2017), and more recently using emergent constraints which relate observed climate trends, variations or other variables to
ECS using an ensemble of models (Caldwell et al., 2018; Cox et al., 2018a). However, debate still rages about the likely range
of ECS (Brown et al., 2018; Bretherton and Caldwell, 2019; Cox et al., 2018b; Gregory et al., 2019), in part because observed
global warming is a rather indirect measure of global warming at equilibrium. On the other hand, TCR is more closely related
to the rate of warming, and therefore ought to be more amenable to constraint by the record of global warming (Bengtsson and
Schwartz, 2013; Gregory and Forster, 2008; Jiménez-de-la Cuesta and Mauritsen, 2019; Tokarska et al., 2020). Nevertheless,
the accepted likely range of TCR has also resisted change (Knutti et al., 2017), for reasons we will discuss in this paper. At the
time of the AR5, the CMIP5 ESMs produced central estimates (mean $\pm$ stdev) of ECS (3.3 $\pm$ 0.7 K) and TCR (1.8 $\pm$ 0.3 K),
that were broadly consistent with these IPCC likely ranges. However, there has been a general drift upwards towards higher
climate sensitivities in the new CMIP6 ESMs, such that more than one third of the new CMIP6 models now have ECS values
over 4.5 K (Forster et al., 2019), and five have TCR values over 2.5 K (Table 1). If the real climate system is similarly sensitive,
the Paris climate targets will be much harder to achieve (Tanaka and O'Neill, 2018).

Therefore some key science- and policy-relevant questions arise:

(a) *Are such high climate sensitivities consistent with the observational record?*

(b) *If so, do the CMIP6 models demand an upward revision to the IPCC likely ranges for climate sensitivity?*

We address these questions in this paper by evaluating the historical simulations of global warming from the CMIP6 models.
In particular, we explore an emergent constraint on TCR based on global warming from 1975 onwards (Jiménez-de-la Cuesta
and Mauritsen, 2019; Tokarska et al., 2020), but using the CMIP6 models and observational data up to 2019.

Emergent constraints are increasingly used to assess future change by exploiting statistical relationships in multimodel en-
sembles between an observable and a variable describing future climate (Cox et al., 2018a; Hall et al., 2019). In the work

presented here, we use the latest CMIP6 multimodel ensemble to define an emergent relationship between historical warming (expressed in terms of GMST, the observable) and TCR (the variable related to future climate). In line with published recommendations (Hall et al., 2019; Klein and Hall, 2015), we check the robustness of the resulting emergent constraint against the CMIP5 ensemble, using exactly the same methodology as for CMIP6. We also follow the suggestion of Hall et al. (2019) in striving to base the emergent constraint on sound physical reasoning.

From physical principles, we expect values of TCR to be very well-correlated with simulated global warming across a model ensemble. By definition, TCR is a measure of warming from a simulation that is driven by an exponential 1.0% per year increase in $CO_2$. Historical global warming has been driven by a qualitative similar forcing, albeit somewhat less rapid. Instead of 1.0%, $CO_2$, the atmospheric CO2 concentration has increased at about 0.5% per year since 2000 (Dlugokenchy and Tans, 2019)), augmented by additional positive radiative forcing from other well-mixed greenhouse gases, and partially offset by the cooling effects of anthropogenic aerosols.

The radiative effects of the rise in greenhouse gas concentrations are relatively well-known (Myhre et al., 2013), and are broadly similar in different ESMs. By contrast, the radiative forcing due to changes in anthropogenic aerosols, especially indirect effects via changes in cloud brightness and lifetime, are poorly constrained (Myhre et al., 2013; Bellouin et al., 2019).

These uncertainties in aerosol forcing have hindered attempts to constrain TCR or ECS from the rate of warming, especially during the pre-1980 period when the burning of sulphurous coal led to increases in $CO_2$ and increases in sulphate aerosols, that went up almost together (Andreae et al., 2005). As a result it has been difficult to distinguish, based purely on the observational record of global warming, between a model with high climate sensitivity and strong aerosol cooling, and a model with low climate sensitivity and weak aerosol cooling (Kiehl, 2007).

In order to minimise the effects of uncertainties in aerosol forcing, we need periods in which aerosol radiative forcing changes relatively little compared to the change in radiative forcing due to $CO_2$ and other well-mixed greenhouse gases. Fortunately, this applies to the decades after 1975 when total aerosol load from global $SO_2$ and $NH_3$ emissions were similar to values over the last decade (Stevens et al., 2017). For this reason, we focus on global warming since 1975. However, we also test the robustness of our conclusions to different start dates (see Figure 5(c)), including the start year of 1970 as used by Jiménez-de-la Cuesta and Mauritsen (2019) (hereafter JM19).

To establish an emergent constraint on ECS, we investigate the appropriate functional form between observed warming and climate sensitivity. Due to the slow response of the ocean, this is not expected to be linear, and using a set of assumptions, JM19 proposed an analytical form based on a two-layer box model. By computing the model parameters directly per model, we investigate the appropriateness of this analytical function, and use it to derive an emergent constraint.

The remainder of this paper is organised as follows: in Section 2 we describe our methodological choices; Section 3 contains the emergent constraints on TCR and ECS and Section 4 contains the discussion and conclusions. More technical details concerning the regression methods are given in the Appendix.

**Table 1.** List of CMIP6 models used in this study and their effective radiative forcing at $CO_2$ doubling $F_{2\times}$, the climate feedback parameter $\lambda$, equilibrium climate sensitivity (ECS) and transient climate response (TCR). Mean values are reported for models with multiple realisations. The values of $F_{2\times}$, ECS and $\lambda$ are computed using the Gregory method (Gregory, 2004). Models above the horizontal line were used in the extended simulations to 2019. Models below the line did not have SSP simulations available at time of writing. Consistently derived values for CMIP5 are displayed in the Supplementary Table 1.

| Centre | Model | $F_{2\times}$ | $\lambda$ | ECS | TCR | n | $\Delta$ T | std($\Delta$ T) |
|---|---|---|---|---|---|---|---|---|
| BCC | BCC-CSM2-MR | 3.01 | 0.98 | 3.07 | 1.59 | 1 | 0.64 | |
| CAMS | CAMS-CSM1-0 | 3.95 | 1.71 | 2.31 | 1.72 | 1 | 0.44 | |
| CAS | FGOALS-f3-L | 3.95 | 1.31 | 3.03 | 2.01 | 1 | 0.70 | |
| CCCma | CanESM5 | 3.63 | 0.64 | 5.66 | 2.66 | 50 | 1.27 | 0.10 |
| CNRM-CERFACS | CNRM-CM6-1 | 3.54 | 0.72 | 4.94 | 2.08 | 10 | 0.73 | 0.11 |
| CNRM-CERFACS | CNRM-ESM2-1 | 3.09 | 0.66 | 4.66 | 1.92 | 5 | 0.65 | 0.15 |
| CSIRO-ARCCSS | ACCESS-CM2 | 3.21 | 0.67 | 4.81 | 2.00 | 1 | 0.77 | |
| CSIRO | ACCESS-ESM1-5 | 2.71 | 0.68 | 3.97 | 1.91 | 3 | 0.84 | 0.10 |
| EC-Earth-Consortium | EC-Earth3-Veg | 3.32 | 0.77 | 4.34 | 2.57 | 2 | 0.97 | 0.23 |
| EC-Earth-Consortium | EC-Earth3 | 3.30 | 0.78 | 4.22 | 2.38 | 10 | 0.72 | 0.16 |
| INM | INM-CM4-8 | 2.61 | 1.42 | 1.84 | 1.32 | 1 | 0.61 | |
| INM | INM-CM5-0 | 2.88 | 1.49 | 1.93 | 1.40 | 1 | 0.55 | |
| IPSL | IPSL-CM6A-LR | 3.32 | 0.72 | 4.63 | 2.32 | 6 | 0.85 | 0.10 |
| MIROC | MIROC-ES2L | | | | 1.55 | 1 | 0.62 | |
| MIROC | MIROC6 | 3.76 | 1.47 | 2.56 | 1.52 | 3 | 0.50 | 0.04 |
| MOHC | HadGEM3-GC31-LL | 3.38 | 0.60 | 5.62 | 2.45 | 4 | 1.07 | 0.19 |
| MOHC | UKESM1-0-LL | 3.56 | 0.66 | 5.41 | 2.72 | 5 | 1.13 | 0.13 |
| MPI-M | MPI-ESM1-2-HR | 3.58 | 1.20 | 2.99 | 1.64 | 2 | 0.65 | 0.07 |
| MRI | MRI-ESM2-0 | 3.36 | 1.07 | 3.14 | 1.56 | 5 | 0.73 | 0.06 |
| NCAR | CESM2-WACCM | 3.08 | 0.63 | 4.90 | 1.92 | 3 | 0.97 | 0.15 |
| NCAR | CESM2 | 3.13 | 0.59 | 5.30 | 2.04 | 3 | 0.82 | 0.01 |
| NCC | NorESM2-LM | 3.06 | 1.13 | 2.69 | 1.46 | 3 | 0.63 | 0.18 |
| NOAA-GFDL | GFDL-CM4 | 2.91 | 0.71 | 4.09 | 1.97 | 1 | 0.86 | |
| NOAA-GFDL | GFDL-ESM4 | 3.51 | 1.31 | 2.68 | 1.53 | 2 | 0.79 | 0.15 |
| NUIST | NESM3 | 3.73 | 0.78 | 4.76 | 2.73 | 2 | 0.93 | 0.17 |
| UA | MCM-UA-1-0 | | | | 1.94 | 1 | 0.81 | |
| **Mean** | | 3.69 | 0.95 | 3.90 | 1.96 | 4.9 | 0.78 | 0.12 |
| **Standard deviation** | | 0.40 | 0.34 | 1.18 | 0.42 | 9.4 | 0.19 | 0.06 |
| AS-RCEC | TaiESM1 | | | | 2.34 | | | |
| BCC | BCC-ESM1 | 3.03 | 0.89 | 3.39 | 1.74 | | | |
| E3SM-Project | E3SM-1-0 | 3.23 | 0.60 | 5.38 | 2.99 | | | |
| NASA-GISS | GISS-E2-1-G | 3.89 | 1.43 | 2.71 | 1.68 | | | |
| NASA-GISS | GISS-E2-1-H | 3.55 | 1.14 | 3.12 | 1.89 | | | |
| MOHC | HadGEM3-GC31-MM | 3.36 | 0.61 | 5.52 | 2.37 | | | |
| MPI-M | MPI-ESM1-2-HR | 3.58 | 1.20 | 2.99 | 1.64 | | | |
| SNU | SAM0-UNICON | 3.83 | 1.02 | 3.76 | 2.25 | | | |
| **Mean CMIP6** | | 3.70 | 0.95 | **3.91** | **2.01** | | | |
| **Standard deviation CMIP6** | | 0.39 | 0.33 | 1.17 | 0.42 | | | |
| **Mean CMIP5** | | 3.58 | 1.06 | **3.31** | **1.79** | | | |
| **Standard deviation CMIP5** | | 0.22 | 0.29 | 0.76 | 0.34 | | | |

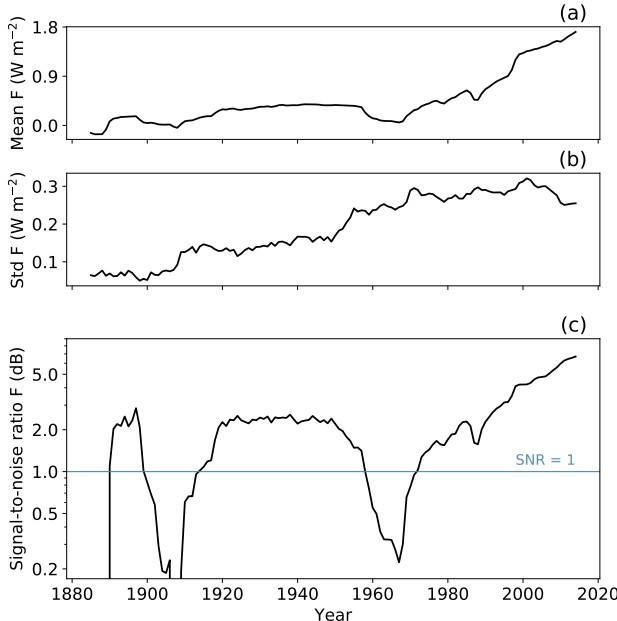

**Figure 1.** Effective radiative forcing over the historical period, calculated from 22 CMIP6 models as $F = \Delta N + \lambda \Delta T$: (a) ensemble mean; (b) ensemble standard deviation; (c) signal-to-noise ratio. Model means are calculated first, and then the ensemble mean is calculated.

## 2 Methodology

### 2.1 Choice of period over which to calculate warming trends

To constrain climate sensitivity using observed warming, we seek a period for which the forcing is relatively similar across models. In order to identify such a period we compute the effective radiative forcing $F$ (ERF) for each model run using

$$F = \Delta N + \lambda \Delta T \tag{1}$$

following Forster et al. (2013). Here $\Delta N$ is the difference in net top of the atmosphere radiative flux and $\Delta T$ is the difference in near-surface temperature, both computed as global annual-mean anomalies relative to the initial state. We calculate the signal-to-noise ratio of $F$ at each time as the model mean $F$ divided by the standard deviation of $F$ across the model ensemble.

Figure 1 shows how the signal to noise ratio of the ERF varies from 1880 to 2010. It is notable that the signal-to-noise ratio increases rapidly from around 1975, as relatively well-known greenhouse gas forcing continues to increase but the uncertain aerosol forcing begins to saturate. We have therefore focused our analysis on the post-1975 warming, but we also performed a sensitivity analysis by varying the start year between 1960 and 2005.

## 2.2 Selection of CMIP6 model runs

We use all currently available CMIP6 models that have control (piControl), historical, a Shared Socioeconomic Pathway simulation (SSP1-2.6, SSP2-4.5, SSP3-7.0 or SSP5-8.5) and one percent $CO_2$ increase per year (1pctCO2) experiments. We extend the historical simulations from 2014 to 2019 using the Shared Socioeconomic Pathways (SSPs) scenario runs. Additional warming over this 5 year period varies very little across the SSPs, so we use SSP2-4.5 as this has the largest number of participating models at the time of writing.

## 2.3 Calculation of model sensitivity

From the 1pctCO2 experiment TCR is determined as the average temperature difference from the corresponding piControl run between 60 to 80 years after the start of the simulation (IPCC, 2013a). ECS is computed using the Gregory method (Gregory, 2004) on the first 150 year of the abrupt-4xCO2 simulations. The values of ECS and TCR that we derived are given in table 1.

## 2.4 Calculation of warming trend

Historical warming (our observable) is found from the historical and SSP simulations using the global annual mean surface air temperature (GMSAT) smoothed with a equally weighted running mean. Some of these models have multiple runs starting from different initial conditions, forcing time series or parameter settings. We use all available runs.

We use smoothed GMSAT to calculate warming. This is to limit the random effect of internal variability on the forced change we wish to constrain. We choose a centred 11-year running mean to remove shorter interannual and mid-term variability from sources such as ENSO, as well as reducing the effect of longer period modes of natural variability. We have tested the robustness of the constraint on TCR to the length of the running mean. It remains relatively invariant past a length of 8 years, suggesting most of the internal variability in GMSAT resides in shorter periods.

Warming $\Delta T$ is calculated as the difference in GMSAT between two periods, typically the years 1975–1985 and 2009–2019 (or equivalently, the difference in smoothed temperature between 2014 and 1980). We have chosen the end year to be 2019 to maximise the chances of discrimination between high and low sensitivity models. As the forcing from $CO_2$ increases with time, the warming in more sensitive models is more likely to diverge from less sensitive ones as we extend the period over which we calculate the trend. Extending to 2019 also allows us to include the most recent observational data and to eliminate possible effects from the warming slowdown between 2000–2012. This slowdown has been attributed to a combination of internal variability and decreased forcing, amongst other things (Medhaug et al., 2017). We assess the impact of the slow-down by comparing emergent constraints derived from time-series truncated to have different end years.

## 2.5 Theoretical basis

### 130    2.5.1    Transient Climate Response (TCR)

Once choices of length of running mean and start and end years for calculation of $\Delta T$ are fixed (our observable), we can fit an emergent relationship between the observable and our values of TCR via linear regression. Linear regression is performed using a hierarchical Bayesian model which can take into account all the different simulations per model: models with more simulations have a better-constrained post-1975 warming. This results in a set of 127 simulations from 26 different models. The

regression method is further described in Appendix A. The choice of linear regression is justified by considering a two-layer energy balance model (Winton et al., 2010; Geoffroy et al., 2013a):

$$C\frac{dT}{dt} = F - \lambda T - \epsilon\gamma(T - T_0)$$
$$C_0\frac{dT_0}{dt} = \gamma(T - T_0). \tag{2}$$

Here $T$ is the top layer temperature anomaly, $T_0$ the deep ocean temperature anomaly, $\lambda$ is the climate feedback parameter, $\epsilon$

is the ocean heat uptake efficacy (reflecting a pattern effect), and $\gamma$ is the ocean heat uptake parameter (Winton et al., 2010). The parameters $C$ and $C_0$ are the heat capacity of the upper ocean and deep ocean, respectively. We will refer to this model as EBM-$\epsilon$, or EBM-1 if $\epsilon$ is set to 1. We follow the approximations in Williamson et al. (2018) and JM19 in assuming no change in deep ocean temperature ($T_0 = 0$), and assuming the upper ocean to be in equilibrium ($dT/dt = 0$). These assumptions are reasonable for timescales larger than a decade, but smaller than a century (see JM19), and lead to the following relationship:

$$\text{TCR} = s\Delta T \tag{3}$$

Here $s$ is a forcing parameter, defined as $F_{2\times}/F$, and $\Delta T$ is the difference in temperature between two periods. For fitting, we include an offset $\eta$, so that $\text{TCR} = s\Delta T + \eta$, allowing for a possible model mis-specification and regression dilution (Hahn, 1977). A hierarchical linear regression was adopted which includes both uncertainty in $\Delta T$ and TCR (see Appendix). The choice of 1975 for the starting period minimises the uncertainty in our estimate of TCR. However, uncertainty is relatively flat

for starting periods between 1975 and 1990. We also investigated the sensitivity of our TCR constraint to the final year, the length of the running mean, the model selection, and the method of regression (see Figure 5).

### 2.5.2    Equilibrium Climate Sensitivity (ECS)

Similarly to the constraint on TCR, we use the warming between 1975–1985 and 2009–2019 to find an emergent constraint on ECS. The relationship between climate sensitivity and observed warming or TCR is not expected to be linear, as a smaller

fraction of equilibrium warming is typically realised in models with high climate sensitive within the first decades of warming (Hansen et al., 1985; Rugenstein et al., 2019). Using Eq. 2, $\text{ECS} = F_{2\times}/\lambda$, and again assuming the upper ocean to be in equilibrium and the deep ocean temperature to not change, TCR and ECS are related via:

$$\text{ECS} = \text{TCR}/(1 - e'\text{TCR}). \tag{4}$$

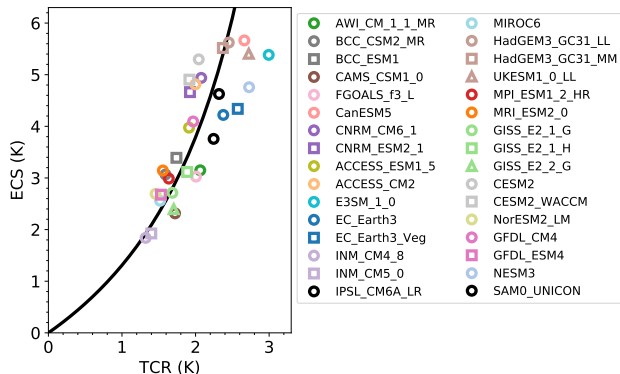

**Figure 2.** Scatter plot of TCR values plotted against ECS values for all CMIP6 models with both available at the time of submission. Models from the same modelling group are plotted with the same colour. Plot markers differentiate models from the same modelling centre. Black line uses the average ocean heat uptake parameters as fitted from the Geoffroy et al. (2013b) procedure, listed in Supplementary Table 3 and related ECS and TCR via: $\text{ECS} = \text{TCR}/(1 - e'\text{TCR})$, with $e' = 0.24$, the model mean.

So that the relationship between ECS and $\Delta T$ ends up as:

$$\text{ECS} = \frac{\Delta T}{s' - e'\Delta T}. \tag{5}$$

The forcing parameter is denoted by $s'$, defined as $\Delta F/F_{2\times}$ and $e'$ is the ocean heat uptake parameter defined as $\epsilon\gamma/F_{2\times}$, The function has an asymptote at $s' - e'\Delta T = 0$, and turns negative for larger $\Delta T$ values. As negative ECS values are unphysical, we modify the equation by keeping ECS at infinity for $\Delta T > s'/e'$. The appearance of negative ECS for high $\Delta T$ is an artefact of the no deep ocean temperature rise assumption: it corresponds to an equilibrium between the heating effect of $F - \lambda\Delta T$, balanced by $-\epsilon\gamma\Delta T$. In reality, this last term cancels completely with $\epsilon\gamma\Delta T_0$ at equilibrium.

To test the validity of these assumptions, we perform two checks. Firstly, by explicitly simulating the two box model, we investigate to what extent the analytical functional form deviates from the true functional form. We are especially interested in the upper region of this functional form, which, if too steep, could lead to an upper estimate of ECS biased high.

Secondly, we fit the ocean heat uptake and forcing parameters for all CMIP6 models, following the two algorithms described in Geoffroy et al. (2013a, b), with slight modifications to ensure solutions exist for all models described in the Supplementary Information.

Using these fitting parameters, we investigate the physical basis of Eq 5 with the EBM-$\epsilon$ and EBM-1 models. If this function derived from the two-box model is a faithful representation, $\Delta T/(s' - e'\Delta T)$ should be better related to ECS with individual model parameters than with the bulk fitted parameter. Figure 2 plots model TCR versus ECS, related via Eq. 4, using the ensemble mean of the fitted ocean parameters.

**Figure 3.** Global mean surface temperature of the 26 CMIP6 models named in Table 1. To avoid visual over-representation, a maximum of ten realisations per model are plotted. An 11-year running mean was used. (a) Temperature anomaly $\Delta T$ using 1880-1910 as reference period. (b) Temperature anomaly $\Delta T$ relative to the 1975–1985 mean.

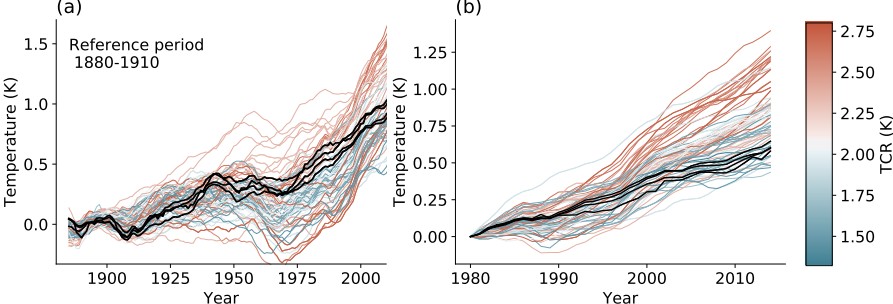

## 3  Results

### 3.1  Transient Climate Response (TCR)

Figure 3a shows the temperature anomaly over the period 1880 to 2019 simulated by 26 different CMIP6 models running a total of 127 simulations smoothed with a 11-year running mean. The reference period in this case is 1880-1910. Model runs have been colour coded by their TCR value, with darker red indicating models with higher TCR, and darker blue indicating lower TCR. Black lines are observational global warming datasets over the same period (Morice et al., 2012; Rohde et al., 2013; Lenssen et al., 2019; Zhang et al., 2019). Models with higher TCR either show large warming at the end of the period, or portray a strong aerosol cooling over the 20th century, particularly visible as a dip around 1960-1970 (notably CNRM-ESM1, UKESM1-0-LL and EC-Earth-Veg). Figure 3b shows the same information for the end of the historical period although the reference period is now chosen to be 1975-1985, after the temperature dip. The positive correlation intuitively expected between TCR and temperature increase $\Delta T$ is much clearer for this time interval.

The $\Delta T$ for each model simulation in Fig. 3b is used for the emergent constraint on TCR in Fig. 4a. Observational warming (black vertical dashed line) is the mean of HadCRUT4 (Morice et al., 2012), Berkeley Earth (Rohde et al., 2013), GISSTEMP4 (Lenssen et al., 2019) and NOAA v5 (Zhang et al., 2019). The 90% observational confidence interval (grey shaded vertical area) is a combination of the observational uncertainty and the internal variability. To avoid double-counting observational uncertainty, the 90% regression confidence interval details the uncertainty of the best estimate of $\Delta T$ versus TCR (see Appendix for details). The models from the previous CMIP5 generation generally fall within the prediction interval of the CMIP6 emergent constraint: the emergent constraint is robust across generations (Klein and Hall, 2015). The best estimate (1.68 K) from this emergent constraint is higher than the best estimate using the larger set of models that have historical simulations up to 2014, but no future scenarios (median: 1.54 K, 5–95% range: 0.76–2.30 K). This can mostly be explained by the fact that 2004-2014 overlaps with the slow-down in surface temperature increase over the 2000-2012 period, but the wider range of models also impacts the regression.

**Figure 4. (a)** Emergent constraint on TCR against historical warming $\Delta T$. $\Delta T$ is calculated from the difference between 1975-1985 and 2009-2019 of a timeseries of GMSAT. Linear regression is performed with all CMIP5 and CMIP6 simulations. Shaded areas indicate a 90% prediction interval (see Appendix A). The vertical dotted line is the mean value of the observations and $y$-axis shows the probability distribution of both generations of ensembles. **(b)** Comparison of probability distributions for the transient climate response using post-1975 warming using CMIP5 and CMIP6 simulations. The probability distribution in the fifth IPCC assessment is not fully specified, so the figure shows a normal distribution with the same likely range as IPCC.

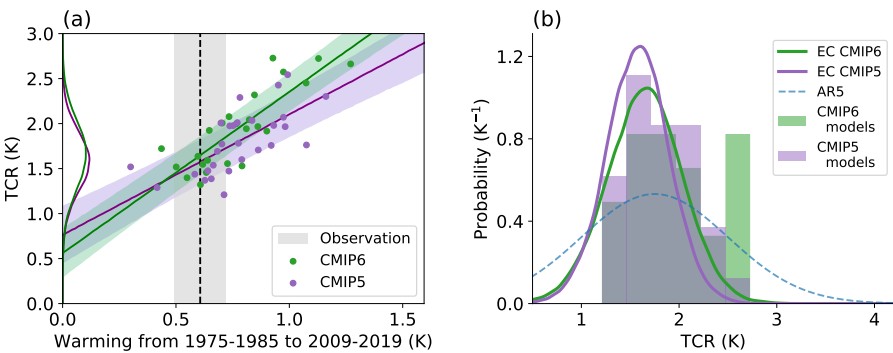

Figure 4b shows the probability density functions (pdf) of TCR derived from the emergent constraint for both CMIP6 and the earlier CMIP5 model ensembles. For comparison, the raw model range in each CMIP is plotted as a histogram, as well as the reported IPCC AR5 likely range (assuming a normal distribution). Both CMIP5 and CMIP6 pdfs are very similar (central estimates differ by 0.1 K) even though CMIP6 contains many more high TCR models. As a continuation of the historical CMIP5 simulation, RCP8.5 is chosen. The tighter constraint in CMIP5 is mostly a consequence of differences in internal variability, which is 42% larger in CMIP6 than in CMIP5, in line with the findings of Parsons et al. (2020).

### 3.1.1 Period selection

Estimates of TCR depend on the final year chosen for the emergent constraint. Uncertainty in the estimate of TCR reduces as time increases and the central estimate converges as shown in Fig. 5a. Later end years are favoured as the signal-to-noise ratio of the net radiative forcing increases monotonically after 1975 (see Figure 1). In the 21st century, the climate impact of volcanoes has been dominated by smaller eruptions (Stocker et al., 2019). The scenarioMIP simulations used for 2015-2019 include a small background forcing from volcanoes (O'Neill et al., 2016). We estimate errors from a potential mismatch between model and real forcing to be relatively small.

To mitigate the effect of internal variability, we use a running mean of GMSAT. Figure 5b shows the likely range of TCR as a function of the length of the running mean. Since we use all available simulations including multiple realisations of the same model in the emergent constraint, the effect of internal variability is already reduced and the length of the running mean on the estimate of TCR is small - the central estimate and the likely range remain relatively invariant past a window length of 8 years.

| Study | Ensemble | Period | Median | 5-95% range | 16-84% range |
|---|---|---|---|---|---|
| Jiménez-de-la Cuesta and Mauritsen (2019) | CMIP5 | 1970 - 2005 | 1.7 K | 1.2 – 2.2 K | |
| Nijsse et al. (2020) | CMIP5 | 1970 - 2005 | 1.7 K | 1.1 – 2.3 K | 1.4 – 2.1 K |
| Tokarska et al. (2020) | CMIP6 | 1981 - 2017 | 1.6 K | | 1.2 – 2.0 K |
| Nijsse et al. (2020) | CMIP6 | 1975 - 2019 | 1.7 K | 1.0 – 2.3 K | 1.3 – 2.1 K |

**Table 2.** Emergent constraint on TCR depending on choices of ensemble and period. Results from JM19 and Tokarska et al. (2020) are also shown for comparison.

Figure 5c shows the effect of the start year on the emergent constraint. Uncertainty in the estimated value of TCR is relatively flat between start years of 1975 and 1990. Uncertainty for start years from 1990 onwards increases until the estimate and the uncertainty revert towards the raw CMIP6 ensemble statistics (no predictive power) at later years.

### 3.1.2 Regression method

When only one realisation per model is used for ordinary least square regression, regression dilution takes place in which the slope is underestimated (Cox et al., 2018b). This has the potential to lead to a slight overestimation of TCR (Fig. 5d), as the observed warming is on the lower end of the model range. JM19 used the average warming for models with multiple simulations. As not all models provide a sufficient amount of simulations, they state that this leads to a minor inflation of the estimation of uncertainty. Although we use a hierarchical Bayesian model as the default (details in Appendix A) we have investigated three other regression methods used in the emergent constraint literature: ordinary least squares (OLS) with only one realisation per model, OLS on the mean warming per model and orthogonal distance regression (Fig. 5d). While the first three give very similar results, orthogonal distance regression gives a somewhat lower estimate of TCR. Orthogonal distance regression assumes that there are both errors in the predictor and in the predictand, which leads to a steeper slope. As our observation lies under the average, a steeper slope results in a smaller mean TCR value. Orthogonal distance regression is known to sometimes overcompensate for errors in the independent variable, for instance in the case the statistical model is not perfectly known; if the model deviates from being a perfectly straight line (Carroll and Ruppert, 1996).

### 3.1.3 Model selection

Model selection can prevent double counting of very similar models (Sanderson et al., 2015; Cox et al., 2018a). As models from the same centre can have very dissimilar climate sensitivities (Chen et al., 2014; Jiménez-de-la Cuesta and Mauritsen, 2019) and sensitivity can change drastically with only small adjustments to parameters (Zhao et al., 2016), we initially use all available models in the CMIP5 and CMIP6 ensemble. Figure 5e shows that this choice does not significantly change the best estimate of the transient response, and that using one model per modelling centre only very slightly increases the variance, even as models from one modelling centre are relative similar (Fig. 2).

**Figure 5.** Robustness of the result to various parameter choices and the choice of regression method. Unless stated differently, start year is 1975, all years up to 2018 are used, and the length of the running mean is 11 years. For comparison, the 5-95% model range range and IPCC range are shown, both assuming normal distributions. **(a)** 5-95% TCR range as a function of the final year (blue line central estimate). **(b):** 5-95% TCR range as a function of length of running mean. **(c)** 5-95% TCR range as a function of start year. **(d)** Pdf of TCR from different regression methods: the hierarchical Bayesian model is compared to three other linear regression methods used in the emergent constraint literature: ordinary least squares (OLS) with only one realisation per model and OLS on the mean warming per model and orthogonal distance regression (ODR). **(e)** Resulting pdfs on TCR from stricter model selection (one model per modelling centre) compared to regression using all models and the IPCC AR5 range.

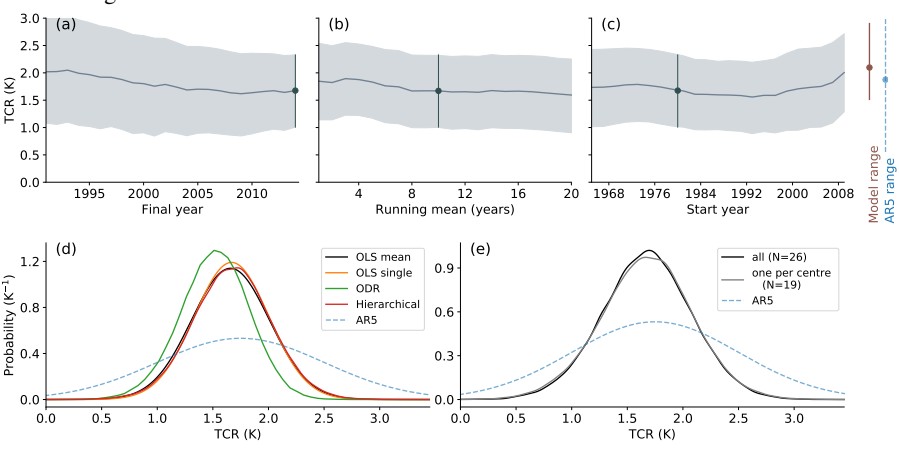

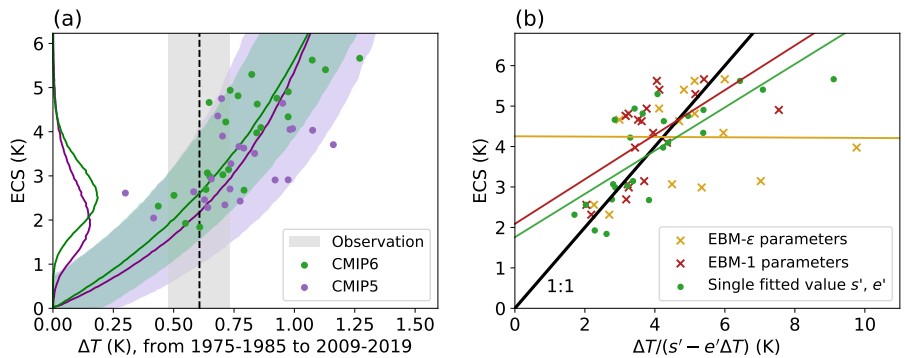

**Figure 6.** a) Emergent constraint for ECS, using the functional form of Eq. 5. The shaded area includes the 5-95% confidence interval. b) Comparison of emergent constraint fitted parameters, with using model values for $s'$ and $e'$. The coloured lines are OLS fits for the three cases, and the black line indicates the 1:1-line. Three values for the EBM-$\epsilon$ model are not shown as their $\Delta T/(s' - e'\Delta T)$ are between 75 and 90 K.

| Ensemble | Median | 5-95% range |
|---|---|---|
| CMIP5 1970-2005 | 2.3 K | 0.7 – 8.4 K |
| CMIP5 hist + RCP85 | 2.2 K | 1.0 – 4.1 K |
| CMIP6 1970-2005 | 2.5 K | 1.0 – 8.6 K |
| CMIP6 historical | 1.9 K | 1.0 – 3.3 K |
| CMIP6 hist + SSP2-4.5 | 2.6 K | 1.5 – 4.0 K |

**Table 3.** Emergent constraint on ECS depending on choice of ensemble and period.

## 3.2 Equilibrium Climate Sensitivity (ECS)

Figure 6a shows the emergent constraint on ECS. For CMIP5, the 5–95% confidence interval lies between 0.96–4.09 K. The constraint is stronger for CMIP6, with the 5–95% confidence interval spanning 1.52–4.03 K. Further results are shown in Table 3.

The results are highly dependent on the time interval chosen. For shorter intervals, the theoretical functional form shows an increased steepness for higher values of $\Delta T$, making it more difficult to constrain. For instance, taking the time period in line with JM19, i.e. 1970–1989 versus 1994–2005, we obtain a 5–95% interval of 0.70–8.41 K for CMIP5, significantly wider than found in JM19, which reported a 5–95% confidence interval of 1.72--4.12 K. The major differences lie in the definition of the theoretical function, where we have cut off the unphysical branch, and a correction of a coding error.

In Fig. 6b the dark green dots represent expected ECS from observed warming (using Eq. 5) and true ECS, using the fitted parameters from Fig. 6a. The light green dots denote the same, but now every model uses its own ocean parameters, $F_{2\times}$ and model forcing computed using Eq. 1. The yellow data shows the expected ECS computed from the EBM-1 model. Full parameter fits for both models are found in Supplementary Tables 2 and 3.

The EBM-$\epsilon$ model performs poorly for large values of the ocean heat uptake efficacy parameter $\epsilon$. Models with $\epsilon$ around 1.8 in particular show an expected ECS far above a realistic range, with one expected ECS reaching a value of 89 K. Eq. 5 is nonlinear and small errors in parameter estimation quickly lead to large errors in ECS. For the EBM-$\epsilon$ model in particular, high internal variability may skew the parameter estimate upwards.

The EBM-1 fit leads to an improved estimation of ECS compared to the Eq. 5 fit in 53% of the cases, whereas the EBM-$\epsilon$ model leads to an improvement in 34% of cases. This pattern in similar in the case only historical models are used, with 66% and 42% improved respectively.

### 3.2.1 Functional form

Explicitly simulating the two-layer model shows that the steepness of the graph is overestimated: assuming no deep ocean temperature rise ($T_0 = 0$) dampens the temperature response of the upper ocean. Geoffroy et al. (2013a) derived an analytical solution to the two-box model of Eq. 2 under the weaker assumption of a linearly increasing forcing, which also showed a less steep increase of ECS with $\Delta T$ for high values of $\Delta T$. This leads the question whether the upper range of ECS is overestimated. In Supplementary Fig. 1, we show this is not the case: by using a decreased ocean heat uptake parameter $e'$

and forcing, the two analytical solutions do overlap, which demonstrates that using the approximated Eq. 5 in the regression should not lead to biased results in the emergent constraint, but simply that the fitted parameters will be slightly different from the model parameters. This also explains why the regression using model parameters in Fig. 6b is not significantly better than using the overall fitted parameters of Fig. 6a.

## 4 Discussion and Conclusion

The emergent constraint found on TCR in this paper is very similar to the one found in JM19 and Tokarska et al. (2020). The most important determinant of the constraint is the periods taken. We have slightly expanded on the amount of models compared to a Tokarska et al. (2020), taking a different period, and we compared further regression choices.

Our best estimate for TCR from the CMIP6 models is 1.68 K, which remains close to the centre of the likely range (1–2.5K) given in the IPCC AR5 (IPCC, 2013b). The emergent constraint on TCR from the CMIP6 models is however strong enough to indicate a much tighter likely range of TCR (16-84%, 1.29–2.05 K).

We find a consistent emergent constraint from the CMIP5 models against observed global warming from 1975 to 2019 (16-84%, 1.27–1.88 K). Furthermore, both of these likely ranges overlap strongly with the emergent constraint on TCR derived by Jiménez-de-la Cuesta and Mauritsen (2019) using a similar method, but only considering global warming from 1970 to 2005 (5-95%, 1.17–2.16 K). In terms of the classification proposed by Hall et al. (2019), we therefore now have a *confirmed* emergent constraint on TCR, with consistency across generations and a sound theoretical framework.

Equilibrium climate sensitivity is likely between 1.9 and 3.4 K (16-84% percentile). This finding strengthens previous evidence that ECS very unlikely above 4.5 K (Cox et al., 2018a; Jiménez-de-la Cuesta and Mauritsen, 2019; Goodwin et al., 2018). For instance, Goodwin et al. (2018) used history matching, a simple emulator, and observations of surface temperature, ocean heat uptake, and carbon fluxes to estimate climate sensitivity and concluded upon a 5-95% range of 2.0 to 4.3 K. Renoult et al. (2020) used a combined emergent constraint of the last glacial maximum and mid-Pliocene Warm Period to constrain ECS to 1.1–3.9 K, with the same best estimate of 2.6 K.

Does the presence of many models with ECS over 4.5 K mean that the CMIP5 generation was better or more useful for understanding climate sensitivity than CMIP6? From the point of view of emergent constraints the answer is clearly no, as model spread helps capture the shape of the emergent relationship.

In the future, we hope that this TCR constraint will become the basis for constraints also on TCRE (transient climate response to emissions), but this will require the inclusion of additional constraints on land and ocean carbon uptake.

However, we are now in a position to answer the questions that we posed in Section 1:

(a) *Are such high climate sensitivities consistent with the observational record?*

No, models with high ECS (>4.5K) and high TCR (>2.5 K) do not appear to be consistent with observed global warming since 1975 (Figure 3b).

(b) *If so, do the CMIP6 models demand an upward revision to the IPCC likely ranges for climate sensitivity?*

**Figure A1.** Schematic of the hierarchical Bayesian model employed. The data layer models a best estimate of historical warming for each model. With this estimate, a regression is performed between historical warming and TCR in the process layer. Using information from both layers and observed warming, a probability density function is estimated for TCR is the final step.

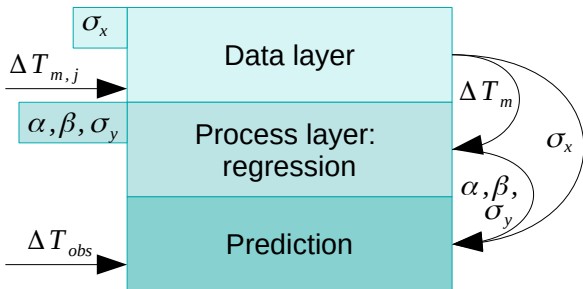

No, instead emergent constraints on TCR (Fig. 4) and ECS (Fig 6) suggest narrower likely ranges for TCR (1.3–2.1 K) and ECS (1.9–3.4 K).

*Code availability.* The code to analyse the data and produce the figures is available upon request to the corresponding author.

*Data availability.* CMIP5 and CMIP6 data can be accessed through ESGF nodes.

## Appendix A: Hierarchical linear regression

To systematically include the information from all model realisations, we use a hierarchical Bayesian model (Sansom, 2014). This model includes two layers: the normal linear regression (process layer) and a layer that computes the expected warming per model from all its initial value realisations (data layer). To include the initial value ensemble, we assume that each model $m$ has a "true" or "best" value for warming over the last decades denoted by $\Delta T_T$. We further assume that every realisation $j$ of a model gives a value of $\Delta T$ that is drawn from a normal distribution with mean $\Delta T_T$ and a standard deviation $\sigma_x$ that is the same across all models. Our hierarchical model consists of two steps: for each model the best estimate of historical warming is computed and with this value a simple linear regression is performed:

$$\Delta T_{m,j} | \Delta T_m, \sigma_x \quad \sim \text{normal}(\Delta T_m, \sigma_x)$$

$$\text{TCR}_m | \alpha, \beta, \sigma_y \quad \sim \text{normal}(\alpha + \beta \Delta T_m, \sigma_y)$$

The probability density function for TCR is then sampled from the observed warming between 1975–1985 and 2009–2019 $\Delta T_{obs}$ using the emergent constraint. The observational uncertainty $\sigma_{obs}$ is taken as the sample standard deviation of the four

observational datasets.

$$\text{TCR}_{pred} = \text{normal}\left(\alpha + \beta\,\text{normal}\left(\Delta T_{obs}, \sqrt{\sigma_x^2 + \sigma_{obs}^2}\right), \sigma_y\right)$$

The second layer corresponds with normal linear regression, while the first layer makes an estimate of the true $\Delta T_m$. Note that especially for models with only few initial value member, the "best" $\Delta T_m$ does not necessarily correspond with the mean value of these ensemble members, but will instead lie closer to the regression line.

As no warming is expected if climate sensitivity were zero, we expect the regression to pass through the intercept and chose a prior for the intercept $\alpha$ of normal(0, 1). Weakly informative priors are chosen for the slope $\beta$, the uncertainty of the regression

$\sigma_y$ and the internal variability $\sigma_x$:

$$\alpha \quad \sim \text{normal}(0,1);$$
$$\beta \quad \sim \text{normal}(2,10);$$
$$\sigma_y \quad \sim \text{half-normal}(0.5,10);$$
$$\sigma_x \quad \sim \text{half-normal}(0.2,0.5);$$

*Author contributions.* All authors contributed towards the design of the study. MSW led on the data collection, FJMMN led on the data analysis with contributions from MSW and PMC. All authors contributed equally to the manuscript.

*Competing interests.* The authors declare no competing interests.

*Acknowledgements.* This work was supported by the European Research Council ECCLES project, grant agreement number 742472 (F.J.M.M.N., P.M.C. and M.S.W.); the EU Horizon 2020 Research Programme CRESCENDO project, grant agreement number 641816 (P.M.C. and
330    M.S.W.). We also acknowledge the World Climate Research Programme's Working Group on Coupled Modelling, which is responsible for CMIP, and we thank the climate modelling groups (listed in Table 1) for producing and making available their model output. We further thank Diego Jiménez-de-la-Cuesta for kindly sharing their code.

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
