# Peer review of "An emergent constraint on Transient Climate Response from simulated historical warming in CMIP6 models"

_Earth System Dynamics, 2019_

## Referee Comment (RC1) · Anonymous Referee #1 · 16 Jan 2020

Review of "An emergent constraint on transient climate response from simulated historical warming in CMIP6 models" by F. J. M. M. Nijsse and co-authors.

In this paper the authors apply a recently proposed emergent constraint on transient climate response (TCR) on a new set of climate models (CMIP6). The emergent constraint uses warming since the 1970's which is a period that has aerosol forcing which doesn't change too much, and so even if there is uncertainty in the absolute magnitude shouldn't affect the warming rate too much. A best estimate TCR of 1.82 K is obtained, which is about 10 percent higher than that found in other studies. These other studies are Jimenez-de-la-Cuesta and Mauritsen (2019), Tokarska et al. (submitted), and

implicitly Winton et al. (2020, JAMES, see their Fig. 14).

It is obviously useful to test a method on new model ensembles, as there have been several cases of emergent constraints found in one ensemble that does not work in another. However, the authors have made a series of choices that are different from the original study which hinders a direct comparison. Again, it is useful that choices regarding the statistics are explored, but it is not currently possible to see whether the shift is related to these new methods or something more fundamental. Other problems were that the authors have not used too many models and some of the writing was less insightful. I suggest the authors undertake major revisions.

Major issues

1) When a study obtains different quantitative estimates compared to previous studies (see above), then I expect to be able to understand why. It is not sufficient to say that this is within the error-bounds because the input data is in principle the same.

2) I found the discussion of ECS somewhat problematic; several detailed comments are provided below. The culmination, however, is at the beginning of section 4, when Figure 4 is discussed, plotting ECS against TCR. From this plot it is claimed that, contrary to earlier studies the post-1970s warming does not constrain ECS. However, the plot uses the posterior TCR to make this claim, not observed warming, and furthermore the authors do not provide a statistical analysis to support the claim. It is furthermore claimed that a straight line is superior to any other more physically based model, which is clearly not right. A physical constraint is that ECS -> 0 as TCR -> 0, and this linear fit is far from crossing the origin. Any curve looks linear if you zoom in far enough.

3) A perhaps somewhat less important point is that the authors first apply smoothing, then average over periods and ensembles, which is effectively the same thing. I mention this because it bothered me that the authors would add an unnecessary layer of complexity, and also because it was unclear what is done with the running-mean smoothing when you approach the end of the time-series in year 2018. For the early

period, nominally 1970-1980, it simply means there is some weighing of years outside the interval, out to 1965-1985 for an 11-year filter. But for the late period, which years are then included? All in all, though, there is no reason to do the smoothing at all, averaging over periods as well as ensemble members is a filter.

Detailed comments

5, Please report what range is given.

23, It is well-known that the TCR/ECS ratio is not a constant, but decreases with ECS (Hansen et al. 1985, Science). We now understand that the ratio is dependent on the feedback, heat uptake coefficient and pattern effects (e.g. Armour 2017).

31, 'climate trends, variability or other observables'

32-34, This sentence left me with an impression that the debate over the value of ECS revolves only around Cox et al. (2018a). Please remove or rewrite.

35, I suggest adding more relevant references, e.g. Gregory and Forster (2008), Otto et al. (2013) and Bengtsson and Schwartz (2013) etc.

40-41, What is this claim based on? Please explain and/or provide references.

42, The questions are also science-relevant, why deprive them to being only policy-relevant?

53, Here, and in several other places, the authors refer to the emergent constraint as theirs ("our constraint"). I suggest rewriting.

58-59, a 1 percent per year increase is also exponential.

Table 1, please add number of simulations and the temperature change.

94-95, Why omit so many years of data, 1980-2008 is not used but contains information as well.

97-99, I didn't understand this, see also major point above.

106, if using 2014 does not significantly change the results, then I suggest to stick with 2014 which would allow including many more models and alleviates concerns that stitching together two experiments could lead to biases (e.g. from missing volcanoes in scenarios).

113, what is "post-aerosol"?

Figure 1, it would seem that more than 13 lines are plotted.

131, this type of information belongs in Methods.

133, I would like to see CMIP5 models tabulated as well.

153, I was confused over this sentence, do the authors mean to refer to 3a instead, and the case where end and start year are so close that there is no signal?

175, I am not sure Rugenstein et al. (2019) said this.

176, likewise, I don't think Jimenez-de-la-Cuesta and Mauritsen (2019) said this.

Appendix A, I struggled to understand this. Would it be possible to provide an illustration of how the method works?

---

## Referee Comment (RC2) · Anonymous Referee #2 · 5 Mar 2020

Review for "An emergent constraint on Transient Climate Response. . ." by F. Nijsse et al. In this paper authors apply the concept of "emergent constraints" to new CMIP6 model data aiming to restrict a possible range of Earth climate system sensitivity to $CO_2$ doubling. The topic of the paper is of considerable importance especially in the light that many of CMIP6 model demonstrate increased sensitivity to $CO_2$ forcing (5K+/2x$CO_2$). The paper fits well within the scope of the journal. I recommend the paper for publishing in general but I think some aspects of the paper should be improved.

General comments:

1. The concept of emergent constraints must be explained much better. Please expand you definition "By definition, we expect..." of emergent constraints on line 55 to be understandable for inexperienced reader .Why TCR has to be correlated with GMST changes across a model ensemble? Models are different, some could have wrong dynamics and incorrect response (Green) function correspondent to $CO_2$ forcing etc. From the paper conclusions it follows that some of the models have wrong TCR while other models are based on the same principles and use more or less the same parameterizations, so why one should believe that TCR/GMST change ratio should be the same for models and for real climate system?

2. Authors must be more careful with the use of definitions.

As far as I understood, the TCR is defined as the change of model/climate system GMST (in K) from equilibrium conditions at the moment of $CO_2$ doubling (1%/year forcing for $CO_2$ only). Then what is TCR in "idealized" conditions?

"Global warming" is a general concept, you cannot relate/correlate it with TCR in K (line 50, line 55 etc).

3. It could be interesting to have CMIP5 model results for comparison on Fig.2a and Fig.4 as well.

4. It should be pointed out that GMST changes are estimated with respect to the non equilibrium state (1970-80 average). Will the green line at Fig.2a cross TSR=0 near the out-of-equilibrium temperature-in-1975 (around -0.4K)?

5. Why 13 models only for CMIP6? Zelinka et al., GRL, 2020 analyzed 27 CMIP6 models...

Special comments:

Lines 20-25. TCR and ECS are introduced for ESMs where they are well defined characteristics for each ESM. On the next line (line 26) paper says that "both TCR and ECS remain uncertain". What do you mean here?

Line 50. Relationship between historical warming (expressed in terms of GMST) and TCR?

Line 55-60. "By definition, we expect...". What "definition" do you mean? What are "idealized" conditions? (Are they somehow different from the ones used in your definition of TCR on lines 20-25)?

Line 80. Could you please provide link to the data.

Line 107-108 Can you illustrate the similarity between aerosol forcing in 1970-80 and 2010-2020?

Line 120. "The major uncertainty....". This sentence falls out of the context.

Line 122. Can you give a number for correlation between TCR and deltaT.

Line 129-131. Move this sentence upward to line 85 (definitions of the table 1)?

Line 200 (Appendix). Appendix does not clarify anything. Either remove or expand it.
* * *

---

## Author Comment (AC1) · 24 Mar 2020

Review of "An emergent constraint on transient climate response from simulated historical warming in CMIP6 models" by F. J. M. M. Nijsse and co-authors. In this paper the authors apply a recently proposed emergent constraint on transient climate response (TCR) on a new set of climate models (CMIP6). The emergent constraint uses warm-

ing since the 1970's which is a period that has aerosol forcing which doesn't change too much, and so even if there is uncertainty in the absolute magnitude shouldn't affect the warming rate too much. A best estimate TCR of 1.82 K is obtained, which is about 10 percent higher than that found in other studies. These other studies are Jimenez-de-la-Cuesta and Mauritsen (2019), Tokarska et al. (submitted), and implicitly Winton et al. (2020, JAMES, see their Fig. 14).

It is obviously useful to test a method on new model ensembles, as there have been several cases of emergent constraints found in one ensemble that does not work in another. However, the authors have made a series of choices that are different from the original study which hinders a direct comparison. Again, it is useful that choices regarding the statistics are explored, but it is not currently possible to see whether the shift is related to these new methods or something more fundamental. Other problems were that the authors have not used too many models and some of the writing was less insightful. I suggest the authors undertake major revisions.

— Response: We thank the reviewer for their thoughtful comments on our paper. Their suggestions with regards to ECS were especially important for improving the paper. The revised paper now contains an additional emergent constraint on ECS.

Major issues

1) When a study obtains different quantitative estimates compared to previous studies (see above), then I expect to be able to understand why. It is not sufficient to say that this is within the error-bounds because the input data is in principle the same.

— Response: We have included further comparison with the Jimenez-de-la-Cuesta and Mauritsen study (henceforth JM19). We have identified several differences: a) JM19 had slightly lower values for TCR compared to the IPCC values. We have now computed TCR values directly (instead of using AR5 values) for CMIP5 to ease comparison with JM19 and CMIP6. Our calculations are very close to the standard IPCC values; b) JM19 compared different periods to those used in our draft paper; c) a

minor programming error was found in the analysis software of JM19; d) JM19 used a different statistical method that assumes error solely in the independent variable. This reduces regression dilution. However, for an observation that is less than the average model warming, this method produces lower values (by about 0.1K). Despite these differences, we get a very similar emergent constraint on TCR if we use a similar method (JM19 had a best estimate of 1.67K, and we get 1,66K using similar methods for CMIP5). We have now included the ODR statistical method in a revised figure 3. We also provide details of our attempts to reproduce JM19 in our revised results section, and have added a table of CMIP5 model values to the appendix.

2) I found the discussion of ECS somewhat problematic; several detailed comments are provided below. The culmination, however, is at the beginning of section 4, when Figure 4 is discussed, plotting ECS against TCR. From this plot it is claimed that, contrary to earlier studies the post-1970s warming does not constrain ECS. However, the plot uses the posterior TCR to make this claim, not observed warming, and furthermore the authors do not provide a statistical analysis to support the claim. It is furthermore claimed that a straight line is superior to any other more physically based model, which is clearly not right. A physical constraint is that ECS -> 0 as TCR -> 0, and this linear fit is far from crossing the origin. Any curve looks linear if you zoom in far enough.

— Response: Our aim in including Figure 4 was to demonstrate that a good constraint on TCR does not imply a good constraint on ECS. However, we accept the reviewer's criticism that our discussion of this point was not well justified by the analysis that we presented. In the revised manuscript, we now attempt to fit the non-linear function between ECS and warming trend, as proposed by JM19. Diego Jimenez de la Cuesta kindly provided us with the python code he used in JM19, which we were able to compare to our own code. We conclude the following: (a) there was a minor coding error in the JM19 code (errors switched between x and y variables), and also some arbitrary adjustments made to compensate for said error; (b) when these are corrected, and we use the warming to 2005 (as JM19), we find no useful constraint on ECS; (c) however,

when we use the warming out to 2019, the corrected code produces a constraint on ECS which is similar to that reported by JM19 (95% ranges of ECS: JM19=1.72-4.12 K; our study=1.76-4.52 K). We have also investigated whether the functional form proposed by JM19 is supported by the relationship between ECS and the warming trend across the models:

ECS = DT/(s' - e'DT).

In order to turn this theoretical relationship into an emergent constraint on ECS, JM19 used the ocean heat uptake parameter e' and the radiative forcing parameter s' as fitting parameters. As these values are highly variable between models, the residual of the DT-ECS emergent relationship should be at least partially explained by model differences in ocean heat uptake and forcing. - if this function is indeed theoretically sound. Yet, DT/(s'-e'DT) correlated better with ECS if the fitted s' and e' are used, instead of model-specific s' and/or e', suggesting that the theory used by JM19 is not fully consistent with the results from the CMIP5 and CMIP6 models. We discuss these issues concerning the relationship between ECS and the warming trend in a revised Section 3.

3) A perhaps somewhat less important point is that the authors first apply smoothing, then average over periods and ensembles, which is effectively the same thing. I mention this because it bothered me that the authors would add an unnecessary layer of complexity, and also because it was unclear what is done with the running-mean smoothing when you approach the end of the time-series in year 2018. For the early period, nominally 1970-1980, it simply means there is some weighing of years outside the interval, out to 1965-1985 for an 11-year filter. But for the late period, which years are then included? All in all, though, there is no reason to do the smoothing at all, averaging over periods as well as ensemble members is a filter.

— Response: In fact, there is only one step of time-averaging (the wording smoothing and running mean referred to the same step), and the years 1965-1970 were not used

at all. We have made changes in the text and figure captions to make this clearer (e.g. not using a central year but instead writing the period out explicitly in the x-label of Fig 2).

Detailed comments 5, Please report what range is given.

— Response: this would make the abstract less readable, as it already contains quite a few numbers. The full range is clear from Table 1, and is now also reiterated in the introduction.

23, It is well-known that the TCR/ECS ratio is not a constant, but decreases with ECS (Hansen et al. 1985, Science). We now understand that the ratio is dependent on the feedback, heat uptake coefficient and pattern effects (e.g. Armour 2017).

— Response: we have removed this rule-of-thumb and added a discussion about the reduced ratio.

31, 'climate trends, variability or other observables

— Response: added

32-34, This sentence left me with an impression that the debate over the value of ECS revolves only around Cox et al. (2018a). Please remove or rewrite

— Response: We removed two of the citations, and added two more recent papers by others.

35, I suggest adding more relevant references, e.g. Gregory and Forster (2008), Otto et al. (2013) and Bengtsson and Schwartz (2013) etc.

— Response: we've added the reference to Gregory and Bengtsson.

40-41, What is this claim based on? Please explain and/or provide references.

— Response. We've added a reference to Tanaka & O'Neill, NCC, (2018).

42, The questions are also science-relevant, why deprive them to being only policyrelevant?

— Response. Added.

53, Here, and in several other places, the authors refer to the emergent constraint as theirs ("our constraint"). I suggest rewriting.

— Response: done

58-59, a 1 percent per year increase is also exponential. Table 1, please add number of simulations and the temperature change.

— Response: We've added exponential before 1% to make this point clearer. We'll add more information to Table 1.

94-95, Why omit so many years of data, 1980-2008 is not used but contains information as well.

— Response: Figure 3 shows a sensitivity to parameter choice, of which one is the running mean. We've extended this to 20 years, and conclude not much extra information is obtained if more years are used.

97-99, I didn't understand this, see also major point above.

— Response: we have clarified this by adjusting the figure captions. We now state the difference between period A and B more clearly.

106, if using 2014 does not significantly change the results, then I suggest to stick with 2014 which would allow including many more models and alleviates concerns that stitching together two experiments could lead to biases (e.g. from missing volcanoes in scenarios).

— Response: With the addition of more models and an extra year, the difference between ending in 2014 and 2019 is more pronounced. There is a trade-off between using a shorter period, that was also dominated by the 'global warming hiatus', and

biases from a discrepancy between real and modelled forcings. We have now investigated the sensitivity to forcings by comparing the various SSPs. A difference of only 1% was found between DT for SSP126 and SSP585.

113, what is "post-aerosol"? Figure 1, it would seem that more than 13 lines are plotted.

— Response: post-aerosol changed into post-1970. In Figure 1, up to ten lines per model were shown. To emphasize this, the sentence stating this has been brought forward in the caption.

131, this type of information belongs in Methods.

— Response: moved.

133, I would like to see CMIP5 models tabulated as well.

— Response: we will do this.

153, I was confused over this sentence, do the authors mean to refer to 3a instead, and the case where end and start year are so close that there is no signal?

— Response: The sentence referred to 3c. However, with more data analysed, we have adjusted the paragraph. The uncertainty is now minimum between 1970 and 1976. The second part of the confusing sentence has been deleted, as uncertainty is not that big anymore with the larger set of models.

175, I am not sure Rugenstein et al. (2019) said this.

— Response: We removed the sentence, and replaced it with a discussion of the nonlinear relationship.

176, likewise, I don't think Jimenez-de-la-Cuesta and Mauritsen (2019) said this.

— Response: Likewise deleted.

Appendix A, I struggled to understand this. Would it be possible to provide an illustration of how the method works?

— Response: we have added an illustration, replaced the pseudo code with equations and simplified the text.

Geoffroy, O., Saint-Martin, D., Bellon, G., Voldoire, A., Olivié, D. J. L., & Tytéca, S. (2013). Transient climate response in a two-layer energy-balance model. Part II: Representation of the efficacy of deep-ocean heat uptake and validation for CMIP5 AOGCMs. Journal of Climate, 26(6), 1859–1876. https://doi.org/10.1175/JCLI-D-12-00196.1

Williamson, M. S., Cox, P. M., & Nijsse, F. J. M. M. (2018). Theoretical foundations of emergent constraints: relationships between climate sensitivity and global temperature variability in conceptual models. Dynamics and Statistics of the Climate System, 3(1). https://doi.org/10.1093/climsys/dzy006
* * *

---

## Author Comment (AC2) · 24 Mar 2020

Reviewer comments are listed in italics below, followed by our responses in normal font.

Anonymous Referee #2

In this paper authors apply the concept of "emergent constraints" to new CMIP6 model data aiming to restrict a possible range of Earth climate system sensitivity to CO2 dou-

bling. The topic of the paper is of considerable importance especially in the light that many of CMIP6 model demonstrate increased sensitivity to CO2 forcing (5K+/2xCO2). The paper fits well within the scope of the journal. I recommend the paper for publishing in general but I think some aspects of the paper should be improved.

General comments:

1. The concept of emergent constraints must be explained much better. Please expand your definition. "By definition, we expect..." of emergent constraints on line 55 to be understandable for inexperienced reader. Why TCR has to be correlated with GMST changes across a model ensemble? Models are different, some could have wrong dynamics and incorrect response (Green) function correspondent to CO2 forcing etc. From the paper conclusions it follows that some of the models have wrong TCR while other models are based on the same principles and use more or less the same param- eterizations, so why one should believe that TCR/GMST change ratio should be the same for models and for real climate system?

— Response: we have added further explanation of the concept and assumptions (see response to comment 2 below).

2. Authors must be more careful with the use of definitions. As far as I understood, the TCR is defined as the change of model/climate system GMST (in K) from equilibrium conditions at the moment of CO2 doubling (1%/year forcing for CO2 only). Then what is TCR in "idealized" conditions? "Global warming" is a general concept, you cannot relate/correlate it with TCR in K (line 50, line 55 etc).

— Response: the paragraph about TCR has been rewritten, making clear that there is only one definition of TCR, and that global warming is defined here as rise in GMST. We further explicitly acknowledge that emergent constraints assume there is no systematic error in the relationship, with a reference to Winkel, Myneni & Brovnik (Earth system dynamics, 2019) and how this emergent constraint is in line with theoretical expectation (JM19).

3. It could be interesting to have CMIP5 model results for comparison on Fig.2a and Fig.4 as well.

—Response: We've added the CMIP5 models to Figure 2a, but Figure 4 became too crowded with both model ensembles.

4. It should be pointed out that GMST changes are estimated with respect to the nonequilibrium state (1970-80 average). Will the green line at Fig. 2a cross TSR=0 near the out-of-equilibrium temperature-in-1975 (around -0.4K)?

—Response: we have adjusted the limits of the figure so that the intercept is visible. The intercept location is highly dependent on the regression method, with those methods assuming the error to be solely in the y-variable (OLS, Hierarchical) getting a positive intercept, while methods assuming similar errors in x and y (orthogonal distance regression), portraying a negative intercept. Per the theoretical foundations of JM19, we expect an approximate zero-intercept in this non-equilibrium regime.

5. Why 13 models only for CMIP6? Zelinka et al., GRL, 2020 analyzed 27 CMIP6 models...

—Response: at the time of submission, there were only 13 models for which all necessary information was available, including future scenarios. Now a larger set of 24 models is available. We have also included the emergent constraint with 31 models that ends in 2014 for which scenarios runs are not required.

Special comments:

Lines 20-25. TCR and ECS are introduced for ESMs where they are well defined characteristics for each ESM. On the next line (line 26) paper says that "both TCR and ECS remain uncertain". What do you mean here?

—Response: We have clarified that the TCR and ECS values that we seek relate to the real climate system. These real-world values are still poorly known, even though ECS and TCR are well-defined for each model.

Line 50. Relationship between historical warming (expressed in terms of GMST) and TCR?

—Response: Added.

Line 55-60. "By definition, we expect...". What "definition" do you mean? What are "idealized" conditions? (Are they somehow different from the ones used in your definition of TCR on lines 20-25)?

—Response: The paragraph has been rewritten. 'By definition' has been changed to 'from physical principles', and it has been made clear we use the normal definition of TCR.

Line 80. Could you please provide link to the data?

—Response: In addition to a reference to the ESGF nodes, we will upload all the code, including the data, to Code Ocean.

Line 107-108. Can you illustrate the similarity between aerosol forcing in 1970-80 and 2010-2020?

—Response: we have added a graph computing the spread in effective radiative forcing in the appendix. This graph shows that the spread is highest in the sixties and early seventies.

Line 120. "The major uncertainty....". This sentence falls out of the context.

—Response: we have removed the sentence and moved the reference to the introduction.

Line 122. Can you give a number for correlation between TCR and deltaT?

—Response: Yes, the correlation is 0.84 for CMIP6, and 0.63 for CMIP5. Added.

Line 129-131. Move this sentence upward to line 85 (definitions of the table 1)?

—Response: done.

Line 200 (Appendix). Appendix does not clarify anything. Either remove or expand it.

—Response: the Appendix has been rewritten completely and a figure has been added for extra clarity. The pseudo-code has been replaced by normal equations for easier understanding.

---

## Referee Report (RR1)

Page 7 must be rewritten as follows (eq.3 is obviously wrong! It must include s, not s prime and be in the form of JM19's Eq.5).

Lines 143-147:

…. We follow the approximations in Williamson et al. (2018) and JM19 in assuming no change in deep ocean temperature, and assuming the upper ocean to be in equilibrium. These assumptions are reasonable for timescales larger than a decade, but smaller than a century (JM19). As a result (see JM19) we have:

$\Delta T_0 = 0$, $\Delta T(\lambda + \varepsilon \gamma) = F$ and $TSR(\lambda + \varepsilon \gamma) = F_{2x}$. Consequently,

$$TSR = F_{2x}/(\lambda + \varepsilon \gamma) = s\Delta T. \qquad (3)$$

Here s is a forcing parameter, defined as $s = F_{2x}/F$. The choice of …..

Lines 155-156:

. …. (Rugenstein et al., 2019). Using (2) and assuming both upper and deep ocean to be in equilibrium we have $\Delta T_0 = \Delta T$ and $ECS = F_{2x}/\lambda$. Expressing $\lambda$ using (3) one can relate TCR and ECS via: …..

Line 159:

Where $s' = F/F_{2x}$ and….

---

## Author Response (AR2)

**Response to reviewer 1.**

**We thank the reviewer for the comments, which have lead to us further clarifying the methods and conclusions of our study. Our responses are below in bold.**

Major issues:

1) Section 2.1 is used to formally argue for using 1975 as a starting year rather than 1970 as used in most other studies. However, based on the description I was unable to understand what is being plotted, but if I am right, I don't think it is relevant. I think what is plotted in panel (b) is the standard deviation of forcing with respect to pre-industrial, given that it rises with time. However, for the purpose of the emergent constraint here, this is completely irrelevant, rather what is interesting as a measure of noise is the standard deviation in the forcing change *after* the given year. Likewise, the signal should be the forcing change afterwards.

**Reply: although we choose a default start year of 1975, in Figure 4c we have also assessed the robustness of our constraint to different start years (including 1970 as used by JM19). In the revised manuscript we point more clearly to this sensitivity study. We have also clarified the caption of Figure 1 as requested.**

2) I remain completely confused as to why the authors apply both a running mean and block averaging. In their reply they state that no data outside the blocks is used. If this is true, then years within the block will have been given different weight in order to have results depend on the running mean length. Also, how is it possible to apply a 20-year running mean (Figure 5b) within a block of only 11 years?`

**Reply: we do not separately "apply both a running mean and block averaging" – these are two different descriptions for the same procedure. We have made this clearer in the revised manuscript, by adding: "*(or equivalently, the difference in smoothed T between 2014 and 1980)".***

3) In linear statistical model used to estimate TCR an offset is assumed (lines 145-147). As for the case of ECS, this offset should be around zero since a world with zero TCR should experience zero warming. Nevertheless, the offset is very large 0.8 or 0.6 K. What is the meaning of this, and why not assume up front that the offset is close to zero?

**Reply: the nonzero offset (regression dilution) is a consequence of minimizing the error in the dependent variable. Using ORD, we don't find an offset significantly different from zero, so we don't attach any physical significance to it. When trying to make a prediction, as we are in this study, it is better to allow a non-zero offset (which implies a 2-parameter rather than a 1-parameter fit to the emergent relationship), to allow for a possible model misspecification (Hahn, G. J. (1977). J. Qual. Techn. 9(2)). We make these issues clearer in our revised manuscript.**

4) I found the discussion of the different regression methods somewhat one sided. The authors argue against Orthogonal distance regression, but why would one assume zero error in the post-1975 warming estimates from models? Intuitively I would think that the relative error there should actually be larger than the error in estimating TCR.

**Reply: we do not assume zero error in the post-1975 warming. Similar to OLS, we minimize the error in the dependent variable. Our methodology for taking account of error in the post-1975 warming estimates is discussed in the Appendix. We now point to this more clearly in both our methods and results sections.**

5) Perhaps not a very major point, but the paper uses many different uncertainty ranges, 16-83, 16-84, 5-95, or even 90 percent confidence with unspecified range. IPCC's 'likely' statement means greater than 66 percent probability, not the 16-84 percentile range. Sometimes these are taken to be the same, but they are not. Practically all studies with which results are compared provide 5-95 percent confidence intervals, and I would recommend that this range is primarily used. I understand that the authors might also want to compare with IPCC, but then they should also state that the 'likely' range is not the same as that provided, but a lower-bound. IPCC can and will add various types of uncertainty not accounted for here and obtain broader likely ranges than found in individual studies.

**Reply: the 16-83% was a typo, which is now corrected. We thank the reviewer for pointing this out. We have also replaced the 90 with 5-95%, and clarified other instances that this is the primary use.**

6) The authors should explain that it is not obvious how to handle the cases of negative ECS with large dT. By removing these cases the error distribution is inevitably skewed and therefore biased towards higher ECS. This is arguably equally unreasonable as keeping the cases in. A reasonable alternative is to remove an equal number of cases from the high end of the error distribution to alleviate the bias.

**Reply: the negative fitted ECS values are an artifact of the assumptions of no deep ocean temperature rise. We have expanded the discussion of negative ECS to explain this point.**

7) I was confused over the discussion of the different model fits (lines 241-259), which doesn't seem to lead to any new conclusions. Perhaps I missed something, though, the authors might want to clarify what they learned from the exercise.

**Reply: we have rewritten the results section to highlight the motivation for the different model fits. We have also expanded the explanation in the methods section by adding (to line 164): 'which, if too steep, could lead to an upper estimate of ECS biased high'.**

Minor or technical issues:

31, This sentence is confusing, to first order the reason the ratio is decreasing is because of deep ocean heat uptake. Pattern effects make this effect stronger.

**Reply: we have included 'deep ocean heat uptake'.**

36, Perhaps specify that AR5 used models and historical warming, rather than 'multiple lines of evidence'.

**Reply: for TCR they also used detection and attribution studies, and reconstructions of RF (see p.1112 AR5 WGI). For conciseness, we prefer the current wording.**

40, The Gregory et al. paper was published 2019.

**Reply: corrected, thank you.**

76, Perhaps reference to Kiehl (2007) would be appropriate.

**Response: done.**

80, Which 'last decade'?

**Response: changed to '*recent*', as the source is from 2017, so doesn't capture 2017-2019.**

105, I guess you also need abrupt4xCO2

**Response: yes, added.**

140, I would suggest to use broadly used nomenclature 'ocean heat uptake efficacy (representing a pattern effect) and \gamma the deep ocean heat uptake parameter'

**Response: done.**

143, what is 'JM19'?

**Response: now defined.**

155, perhaps refer to Hansen et al. 1985 for unrealised warming?

**Response: done.**

179-180, These specific models can not be distinguished in the plot.

**Response: distinguishing individual models in the plot would make it overly complicated, in our view.**

181, which 'dip'?

**Response: we have added '*temperature*' to be more specific (refers to previous sentence).**

185-186, how was the confidence interval calculated?

**Response**: **this is explained in the Appendix, which is now more explicitly pointed-to from the main manuscript.**

210-211, It seems to me that the uncertainty increases (surprisingly) little with increasing start year (Fig. 5c). Instead it is the median that increases. Also, there is no reason the uncertainty range should revert to the CMIP6 range, it can be much broader if things are done correctly: Imagine an ensemble with very little spread in TCR, then the uncertainty in the relationship should be very large and the posterior spread of TCR too.

**Reply: as a consequence of using linear regression, the uncertainty range does revert to the range. In the case of low spread in TCR, the slope parameter will get close to zero, and the uncertainty around the slope will encompass the entire ensemble spread. No matter the observation, the result will revert to the ensemble statistics. Only (statistical) model uncertainty can lead to larger spread.**

Figure 5c, which 'Model range'?

**Response: we have included an explanation in the caption: '*For comparison, the 5-95% model range and IPCC range are shown, both assuming normal distributions*'.**

Figure 6b, greens are hard to distinguish, they look the same in print. Also yellow is a poor choice. There are many other colours to choose from.

**Response: we changed the colours; yellow to light brown, light green to dark red. We also changed the markers for the alternative models to crosses, to further emphasize that the dark green dots correspond to the dark green dots on the left-hand panel.**

279, delete 'tests'

**Response: done.**

290, I believe this should be 1.9-3.4 K

**Response: corrected**.

**Response to reviewer 2.**

**We thank the reviewer for their review, who prevented a silly mistake from being printed and whose suggestions mathematics more transparent. Answers in bold.**

Page 7 must be rewritten as follows (eq.3 is obviously wrong! It must include s, not s prime and be in the form of JM19's Eq.5).

**Response: we have corrected eq.3**

Lines 143-147:

…. We follow the approximations in Williamson et al. (2018) and JM19 in assuming no change in deep ocean temperature, and assuming the upper ocean to be in equilibrium. These assumptions are reasonable for timescales larger than a decade, but smaller than a century (JM19). As a result (see JM19) we have:

$\Delta T0 = 0$, $\Delta T(\lambda + \varepsilon \gamma) = F$ and $TSR(\lambda + \varepsilon \gamma) = F2x$. Consequently,

$TSR = F2x/(\lambda + \varepsilon \gamma) = s\Delta T$. (3)

Here s is a forcing parameter, defined as $s = F2x/F$. The choice of …..

**Response: we have included the extra equations in a slightly different wording.**

Lines 155-156:

. …. (Rugenstein et al., 2019). Using (2) and assuming both upper and deep ocean to be in equilibrium we have $\Delta T0 = \Delta T$ and $ECS = F2x/\lambda$. Expressing $\lambda$ using (3) one can relate TCR and ECS via:
…..

Line 159:

Where $s' = F/F2x$ and….

**Response: we have now included the definition of ECS, the reference to eq (2) and reiterated the assumptions.**

[revised manuscript text omitted]